# LIGS: Learnable Intrinsic-Reward Generation Selection for Multi-Agent Learning

**David Mguni**[1,*]**Taher Jafferjee**[1]**, Jianhong Wang**[2]**, Oliver Slumbers**[1,5]**, Nicolas Perez-Nieves**[2]**,
**Feifei Tong**[1]**, Li Yang**[3]**, Jiangcheng Zhu**[1]**, Yaodong Yang**[4]**, Jun Wang**[5]
[1]Huawei Technologies, [2]Imperial College London, [3]Shanghaitech University,
[4]Institute for AI, Peking University & BIGAI, [5]University College London

## Abstract

Efficient exploration is important for reinforcement learners to achieve high rewards. In multi-agent systems, *coordinated* exploration and behaviour is critical for agents to jointly achieve optimal outcomes. In this paper, we introduce a new general framework for improving coordination and performance of multi-agent reinforcement learners (MARL). Our framework, named Learnable Intrinsic-Reward Generation Selection algorithm (LIGS) introduces an adaptive learner, Generator that observes the agents and learns to construct intrinsic rewards online that coordinate the agents' joint exploration and joint behaviour. Using a novel combination of MARL and switching controls, LIGS determines the best states to learn to add intrinsic rewards which leads to a highly efficient learning process. LIGS can subdivide complex tasks making them easier to solve and enables systems of MARL agents to quickly solve environments with sparse rewards. LIGS can seamlessly adopt existing MARL algorithms and, our theory shows that it ensures convergence to policies that deliver higher system performance. We demonstrate its superior performance in challenging tasks in Foraging and StarCraft II.

## 1 Introduction

Cooperative multi-agent reinforcement learning (MARL) has emerged as a powerful tool to enable autonomous agents to solve various tasks such as autonomous driving (Zhou et al., 2020b), ride-sharing (Li et al., 2019), gaming AIs (Peng et al., 2017a), power networks (Wang et al., 2021a; Qiu et al., 2021) and swarm intelligence (Mguni et al., 2018; Yang et al., 2017). In multi-agent systems (MAS), maximising system performance often requires agents to coordinate during exploration and learn coordinated joint actions. However, in many MAS, the reward signal provided by the environment is not sufficient to guide the agents towards coordinated behaviour (Matignon et al., 2012). Consequently, relying on solely the individual rewards received by the agents may not lead to optimal outcomes (Mguni et al., 2019). This problem is exacerbated by the fact that MAS can have many stable points some of which lead to arbitrarily bad outcomes (Roughgarden & Tardos, 2007).

As in single agent RL, in MARL inefficient exploration can dramatically decrease sample efficiency. In MAS, a major challenge is how to overcome sample inefficiency from poorly *coordinated exploration*. Unlike single agent RL, in MARL, the collective of agents is typically required to coordinate its exploration to find their optimal joint policies[1]. A second issue is that in many MAS settings of interest, such as video games and physical tasks, rich informative signals of the agents' *joint* performance are not readily available (Hosu & Rebedea, 2016). For example, in StarCraft Micromanagement (Samvelyan et al., 2019), the sparse reward alone (win, lose) gives insufficient information to guide agents toward their optimal joint policy. Consequently, MARL requires large numbers of samples producing a great need for MARL methods that can solve such problems efficiently.

To aid coordinated learning, algorithms such as QMIX (Rashid et al., 2018), MF-Q (Yang et al., 2018), Q-DPP (Yang et al., 2020), COMA (Foerster et al., 2018) and SQDDPG (Wang et al., 2020c), so-called centralised critic and decentralised execution (CT-DE) methods use a centralised critic whose

---

*Correspondence to davidmguni@hotmail.com.

[1]Unlike single agent RL, MARL exploration issues cannot be mitigated by adjusting exploration rates or policy variances (Mahajan et al., 2019).

role is to estimate the agents' expected returns. The critic makes use of all available information generated by the system, specifically the global state and the joint action (Peng et al., 2017b). To enable effective CT-DE, it is critical that the joint greedy action should be equivalent to the collection of individual greedy actions of agents, which is called the IGM (Individual-Global-Max) principle (Son et al., 2019). CT-DE methods are however, prone to convergence to suboptimal joint policies (Wang et al., 2020a) and suffer from variance issues for gradient estimation (Kuba et al., 2021). Existing value factorisations, e.g. QMIX and VDN (Sunehag et al., 2017) cannot ensure an exact guarantee of IGM consistency (Wang et al., 2020b). Moreover, CT-DE methods such as QMIX require a monotonicity condition which is violated in scenarios where multiple agents must coordinate but are penalised if only a subset of them do so (see Exp. 2, Sec. 6.1).

To tackle these issues, in this paper we introduce a new MARL framework, LIGS that constructs intrinsic rewards online which guide MARL learners towards their optimal joint policy. LIGS involves an *adaptive* intrinsic reward agent, the Generator that selects intrinsic rewards to add according to the history of visited states and the agents' joint actions. The Generator adaptively guides the agents' exploration and behaviour towards coordination and maximal joint performance. A pivotal feature of LIGS is the novel combination of RL and *switching controls* (Mguni, 2018) which enables it to determine the best set of states to learn to add intrinsic rewards while disregarding less useful states. This enables the Generator to quickly learn how to set intrinsic rewards that guide the agents during their learning process. Moreover, the intrinsic rewards added by the Generator can significantly deviate from the environment rewards. This enables LIGS to both promote complex *joint exploration* patterns and decompose difficult tasks. Despite this flexibility, special features within LIGS ensure the underlying optimal policies are preserved so that the agents learn to solve the task at hand.

Overall, LIGS has several advantages:
• LIGS has the freedom to introduce rewards that vastly deviate from the environment rewards. With this, LIGS promotes *coordinated exploration* (i.e. visiting unplayed state-joint actions) among the agents enabling them to find joint policies that maximise the system rewards and generates intrinsic rewards to aid solving sparse reward MAS (see Experiment 1 in Sec. 6.1).
• LIGS selects which best states to add intrinsic rewards *adaptively* in response to the agents' behaviour while the agents learn leading to an efficient learning process (see *Investigations* in Sec. 6.1).
• LIGS's intrinsic rewards preserve the agents' optimal joint policy and ensure that the total *environment* return is (weakly) increased (see Sec. 5).

To enable the framework to perform successfully, we overcome several challenges: **i)** Firstly, constructing an intrinsic reward can change the underlying problem leading to the agents solving irrelevant tasks (Mannion et al., 2017). We resolve this by endowing the intrinsic reward function with special form which both allows a rich spread of intrinsic rewards while preserving the optimal policy. **ii)** Secondly, introducing intrinsic reward functions can *worsen* the agents' performance (Devlin & Kudenko, 2011) and doing so *while training* can lead to convergence issues. We prove LIGS leads to better performing policies and that LIGS's learning process converges and preserves the MARL learners' convergence properties. **iii)** Lastly, adding an agent Generator with its own goal leads to a Markov game (MG) with $N + 1$ agents (Fudenberg & Tirole, 1991). Tractable methods for solving MGs are extremely rare with convergence only in special cases (Yang & Wang, 2020). Nevertheless, using a special set of features in LIGS's design, we prove LIGS converges to a solution in which it learns an intrinsic reward function that improves the agents' performance.

## 2  RELATED WORK

**Reward shaping** (Harutyunyan et al., 2015; Mguni et al., 2021) is a technique which aims to alleviate the problem of sparse and uninformative rewards by supplementing the agent's reward with a prefixed term $F$. In Ng et al. (1999) it was established that adding a *shaping reward function* of the form $F(s_{t+1}, s_t) = \gamma\phi(s_{t+1}) - \phi(s_t)$ preserves the optimal policy and in some cases can aid learning. RS has been extended to MAS (Devlin et al., 2011; Mannion et al., 2018; Devlin & Kudenko, 2011; 2012; 2016; Sadeghlou et al., 2014) in which it is used to promote convergence to efficient social welfare outcomes. Poor choices of $F$ in a MAS can slow the learning process and can induce convergence to poor system performance (Devlin & Kudenko, 2011). In MARL, the question of which shaping function to use remains unaddressed. Typically, RS algorithms rely on hand-crafted shaping reward functions that are constructed using domain knowledge, contrary to the goal of

autonomous learning (Devlin & Kudenko, 2011). As we later describe LIGS, which successfully *learns* an instrinsic reward function $F$, uses a similar form as PBRS however, $F$ is now augmented to include the actions of another RL agent to learn the intrinsic rewards online. In Du et al. (2019) an approach towards learning intrinsic rewards is proposed in which a parameterised intrinsic reward is learned using a bilevel approach through a centralised critic. In Wang et al. (2021b), a parameterised intrinsic reward is learned by a corpus, then the trained intrinsic reward is frozen on parameters and used to assist the training of a single-agent policy for generating the dialogues. Loosely related are single-agent methods (Zheng et al., 2018; Dilokthanakul et al., 2019; Kulkarni et al., 2016; Pathak et al., 2017) which, in general, introduce heuristic terms to generate intrinsic rewards.

**Multi-agent exploration methods** seek to promote coordinated exploration among MARL learners. Mahajan et al. (2019) proposed a hybridisation of value and policy-based methods that uses mutual information to learn a diverse set of behaviours between agents. Though this approach promotes coordinated exploration, it does not encourage exploration of novel states. Other approaches to promote exploration in MARL while assuming aspects of the environment are known in advance and agents can perform perfect communication between themselves (Viseras et al., 2016). Similarly, to promote coordinated exploration in partially observable settings, Pesce & Montana (2020) proposed end-to-end learning of a communication protocol through a memory device. In general, exploration-based methods provide no performance guarantees nor do they ensure the optimal policy. Moreover, many employ heuristics that naively reward exploration to unvisited states without consideration of the environment reward. This can lead to spurious objectives being maximised.

Within these categories, closest to our work is the intrinsic reward approach in Du et al. (2019). There, the agents' policies and intrinsic rewards are learned with a bilevel approach. In contrast, LIGS performs these operations *concurrently* leading to a fast and efficient procedure. A crucial point of distinction is that in LIGS, the intrinsic rewards are constructed by an RL agent (Generator) with its own reward function. Consequently, LIGS can generate complex patterns of intrinsic rewards, encourage *joint exploration*. Additionally, LIGS learns intrinsic rewards only at relevant states, this confers high computational efficiency. Lastly, unlike exploration-based methods e.g., Mahajan et al. (2019), LIGS ensures preservation of the agents' joint optimal policy for the task.

## 3 PRELIMINARIES

A fully cooperative MAS is modelled by a decentralised-Markov decision process (Dec-MDP) (Deng et al., 2021). A Dec-MDP is an augmented MDP involving a set of $N \geq 2$ agents denoted by $\mathcal{N}$ that independently decide actions to take which they do so simultaneously over many rounds. Formally, a dec-MDP is a tuple $\mathfrak{M} = \langle \mathcal{N}, \mathcal{S}, (\mathcal{A}_i)_{i \in \mathcal{N}}, P, R, \gamma \rangle$ where $\mathcal{S}$ is the finite set of states, $\mathcal{A}_i$ is an action set for agent $i \in \mathcal{N}$ and $R : \mathcal{S} \times \mathcal{A} \to \mathcal{P}(D)$ is the reward function that all agents jointly seek to maximise where $D$ is a compact subset of $\mathbb{R}$ and lastly, $P : \mathcal{S} \times \mathcal{A} \times \mathcal{S} \to [0, 1]$ is the probability function describing the system dynamics where $\mathcal{A} := \times_{i=1}^{N} \mathcal{A}_i$. Each agent $i \in \mathcal{N}$ uses a *Markov policy* $\pi_i : \mathcal{S} \times \mathcal{A}_i \to [0, 1]$ to select its actions. At each time $t \in 0, 1, \ldots$, the system is in state $s_t \in \mathcal{S}$ and each agent $i \in \mathcal{N}$ takes an action $a_t^i \in \mathcal{A}_i$. The *joint action* $\boldsymbol{a}_t = (a_t^1, \ldots, a_t^N) \in \mathcal{A}$ produces an immediate reward $r_i \sim R(s_t, \boldsymbol{a}_t)$ for agent $i \in \mathcal{N}$ and influences the next-state transition which is chosen according to $P$. The goal of each agent $i$ is to maximise its expected returns measured by its value function $v^{\pi^i, \pi^{-i}}(s) = \mathbb{E}_{\pi^i, \pi^{-i}} \left[ \sum_{t=0}^{\infty} \gamma^t R(s_t, \boldsymbol{a}_t) \right]$, where $\Pi_i$ is a compact Markov policy space and $-i$ denotes the tuple of agents excluding agent $i$.

Intrinsic rewards can strongly induce more efficient learning (and can promote convergence to higher performing policies) (Devlin & Kudenko, 2011). We tackle the problem of how to *learn* intrinsic rewards produced by a function $F$ that leads to the agents learning policies that jointly maximise the system performance (through coordinated learning). Determining this function is a significant challenge since poor choices can hinder learning and the concurrency of multiple learning processes presents potential convergence issues in a system already populated by multiple learners (Zinkevich et al., 2006). Additionally, we require that the method preserves the optimal joint policy.

## 4 THE LIGS FRAMEWORK

To tackle the challenges described above, we introduce Generator an *adaptive* agent with its own objective that determines the best intrinsic rewards to give to the agents at each state. Using

observations of the joint actions played by the $N$ agents, the goal of the Generator is to construct intrinsic rewards to coordinate exploration and guide the agents towards learning joint policies that maximise their shared rewards. To do this, the Generator learns how to choose the values of an intrinsic reward function $F^{\boldsymbol{\theta}}$ at each state. Simultaneously, the $N$ agents perform actions to maximise their rewards using their individual policies. The objective for each agent $i \in \{1, \ldots, N\}$ is given by:

$$v^{\pi^i, \pi^{-i}, g}(s) = \mathbb{E}\left[\sum_{t=0}^{\infty} \gamma^t \left(R + F^{\boldsymbol{\theta}}\right) \Big| s_0 = s\right],$$

where $\boldsymbol{\theta}$ is determined by the Generator using the policy $g : \mathcal{S} \times \Theta \to [0, 1]$ and $\Theta \subset \mathbb{R}^p$ is the Generator's action set. The intrinsic reward function is given by $F^{\boldsymbol{\theta}}(\cdot) \equiv \theta_t^c - \gamma^{-1}\theta_{t-1}^c$ where $\theta_t^c \sim g$ is the action chosen by the Generator and $\theta_t^c \equiv 0, \forall t < 0$. $\Theta$ can be a set of integers $\{1, \ldots, K\}$). Therefore, the Generator determines the output of $F^{\boldsymbol{\theta}}$ (which it does through its choice of $\theta^c$). With this, the Generator constructs intrinsic rewards that are tailored for the specific setting.

LIGS freely adopts any MARL algorithm for the $N$ agents (see Sec. 10 in the Supp. Material). The transition probability $P : \mathcal{S} \times \mathcal{A} \times \mathcal{S} \to [0, 1]$ takes the state and *only* the actions of the $N$ agents as inputs. Note that unlike reward-shaping methods e.g. (Ng et al., 1999), the function $F$ now contains action terms $\theta^c$ which are chosen by the Generator which enables the intrinsic reward function to be learned online. The presence of the action $\theta^c$ term may spoil the policy invariance result in Ng et al. (1999). We however prove a policy invariance result (Prop. 1) analogous to that in Ng et al. (1999) which shows LIGS preserves the optimal policy of $\mathfrak{M}$. The Generator is an RL agent whose objective takes into account the history of states and $N$ agents' joint actions. The Generator's objective is:

$$v_c^{\boldsymbol{\pi}, g}(s) = \mathbb{E}_{\boldsymbol{\pi}, g}\left[\sum_{t=0}^{\infty} \gamma^t \left(R^{\boldsymbol{\theta}}(s_t, \boldsymbol{a}_t) + L(s_t, \boldsymbol{a}_t)\right) \Big| s_0 = s\right], \quad \forall s \in \mathcal{S}. \tag{1}$$

where $R^{\boldsymbol{\theta}}(s, \boldsymbol{a}) := R(s, \boldsymbol{a}) + F^{\boldsymbol{\theta}}$. The objective encodes Generator's agenda, namely to maximise the agents' expected return. Therefore, using its intrinsic rewards, the Generator seeks to guide the set of agents toward optimal joint trajectories (potentially away from suboptimal trajectories, c.f. Experiment 2) and enables the agents to learn faster (c.f. StarCraft experiments in Sec. 6). Lastly, $L : \mathcal{S} \times \mathcal{A} \to \mathbb{R}$ rewards Generator when the agents jointly visit novel state-joint-action tuples and tends to 0 as the tuples are revisited. We later prove that with this objective, the Generator's optimal policy (for constructing the intrinsic rewards) maximises the expected (extrinsic) return (Prop. 1).

Since the Generator has its own (distinct) objective, the resulting setup is an MG, $\mathcal{G} = \langle \mathcal{N} \times \{c\}, \mathcal{S}, (\mathcal{A}_i)_{i \in \mathcal{N}}, \Theta, P, R^{\boldsymbol{\theta}}, R_c, \gamma \rangle$ where the new elements are $\{c\}$, the Generator agent, $R^{\boldsymbol{\theta}} := R + F^{\boldsymbol{\theta}}$, the new team reward function which contains the intrinsic reward $F^{\boldsymbol{\theta}}$, $R_c : \mathcal{S} \times \mathcal{A} \times \Theta \to \mathbb{R}$, the one-step reward for the Generator (we give the details of this later).

**Switching Control Mechanism**

So far the Generator's problem involves learning to construct intrinsic rewards at *every* state which can be computationally expensive. We now introduce an important feature which allows LIGS to learn the best intrinsic reward only in a subset of states in which intrinsic rewards are most useful. This is in contrast to the problem tackled by the $N$ agents who must compute their optimal actions at all states. To achieve this, we now replace the Generator's policy space with a form of policies known as *switching controls*. These policies enable Generator to decide at which states to learn the value of intrinsic rewards. This enables the Generator to learn quickly both where to add intrinsic rewards and the magnitudes that improve performance since the Generator's magnitude optimisations are performed only at a subset of states. Crucially, with this the Generator can learn its policy rapidly enabling it to guide the agents toward coordination and higher performing policies while they train.

At each state, the Generator first makes a *binary decision* to decide to *switch on* its $F$ for agent $i \in \mathcal{N}$ using a switch $I_t$ which takes values in $\{0, 1\}$. Crucially, now the Generator is tasked with learning how to construct the $N$ agents' intrinsic rewards *only* at states that are important for guiding the agents to their joint optimal policy. Both the decision to activate the function $F$ and its magnitudes is determined by the Generator. With this, the agent $i \in \mathcal{N}$ objective becomes:

$$v^{\boldsymbol{\pi}, g}(s_0, I_0) = \mathbb{E}\left[\sum_{t=0}^{\infty} \gamma^t \left\{R + F^{\boldsymbol{\theta}} \cdot I_t\right\}\right], \forall (s_0, I_0) \in \mathcal{S} \times \{0, 1\}, \tag{2}$$

where $I_{\tau_{k+1}} = 1 - I_{\tau_k}$, which is the switch for $F$ which is $0$ or $1$ and $\{\tau_k\}_{k>0}$ are times that a switch takes place[2] so for example if the switch is first turned on at the state $s_5$ then turned off at $s_7$, then $\tau_1 = 5$ and $\tau_2 = 7$ (we will shortly describe these in more detail). At any state, the decision to turn on $I$ is decided by a (categorical) policy $\mathfrak{g}_c : \mathcal{S} \to \{0, 1\}$ which acts according to Generator's objective. In particular, first, the Generator makes an observation of the state $s_k \in \mathcal{S}$ and the joint action $\boldsymbol{a}_k$ and using $\mathfrak{g}_c$, the Generator decides whether or not to activate the policy $g$ to provide an intrinsic reward whose value is determined by $\theta_k^c \sim g$. With this it can be seen the sequence of times $\{\tau_k\}$ is $\tau_k = \inf\{t > \tau_{k-1} | s_t \in \mathcal{S}, \mathfrak{g}_c(s_t) = 1\}$ so the switching times. $\{\tau_k\}$ *are **rules** that depend on the state.* Therefore, by learning an optimal $\mathfrak{g}_c$, the Generator learns the useful states to switch on $F$.

To induce the Generator to selectively choose when to switch on the additional rewards, each switch activation incurs a fixed cost for the Generator. In this case, the objective for the Generator is:

$$v_c^{\boldsymbol{\pi}, g}(s_0, I_0) = \mathbb{E}_{\boldsymbol{\pi}, g}\left[\sum_{t=0}^{\infty} \gamma^t \left( R^{\boldsymbol{\theta}}(s_t, \boldsymbol{a}_t) - \sum_{k \geq 1} \delta_{\tau_{2k-1}}^t + L(s_t, \boldsymbol{a}_t) \right)\right], \tag{3}$$

where the Kronecker-delta function $\delta_{\tau_{2k-1}}^t$ which is 1 whenever $t = \tau_{2k-1}$ and 0 otherwise imposes a cost for each switch activation. The cost has two effects: first, it reduces the computational complexity of the Generator's problem since the Generator now determines *subregions* of $\mathcal{S}$ it should learn the values of $F$. Second, it ensures the *information-gain* from encouraging the agents to explore state-action tuples is sufficiently high to merit activating a stream of intrinsic rewards. We set $\tau_0 \equiv 0$, $\theta_{\tau_k} \equiv 0, \forall k \in \mathbb{N}$ ($\theta_{\tau_k+1}, \ldots, \theta_{\tau_{k+1}-1}$ remain non-zero), $\theta_k^c \equiv 0 \ \ \forall k \leq 0$ and denote by $I(t) \equiv I_t$.

### Discussion on Computational Aspect

The switching controls mechanism results in a framework in which the problem facing the Generator has a markedly reduced decision space in comparison to the agent's problem (though the agents share the same experiences). Crucially, the Generator must compute optimal intrinsic rewards at only a subset of states which are chosen by $\mathfrak{g}_c$. Moreover, the decision space for the switching policy $\mathfrak{g}_c$ is $\mathcal{S} \times \{0, 1\}$ i.e at each state it makes a binary decision. Consequently, the learning process for $\mathfrak{g}_c$ is much quicker than the agents' policies which must optimise over the decision space $|\mathcal{S}||\mathcal{A}|$ (choosing an action at every state). This results in the Generator rapidly learning its optimal policies (relative to the agent) in turn, enabling the Generator to guide the agents towards its optimal

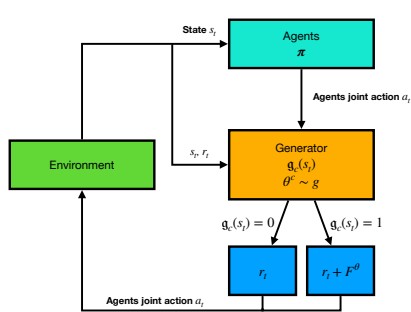

Figure 1: Schematic of the LIGS framework.

policy during its learning phase. Also, in our experiments, we chose the size of the action set for the Generator, $\Theta$ to be a singleton resulting in a decision space of size $|\mathcal{S}| \times \{0, 1\}$ for the entire problem facing the Generator. We later show that this choice leads to improved performance while removing the free parameter of the dimensionality of the Generator's action set.

### Summary of Events

At a time $t \in 0, 1, \ldots$
1. The $N$ agents makes an observation of the state $s_t \in \mathcal{S}$.
2. The $N$ agents perform a joint action $\boldsymbol{a}_t = (a_t^1, \ldots, a_t^N)$ sampled from $\boldsymbol{\pi} = (\pi^1, \ldots, \pi^N)$.
3. The Generator makes an observation of $s_t$ and $\boldsymbol{a}_t$ and draws samples from its polices $(\mathfrak{g}_c, g)$.
4. If $\mathfrak{g}_c(s_t) = 0$:
   ○ Each agent $i \in \mathcal{N}$ receives a reward $r_i \sim R(s_t, \boldsymbol{a}_t)$ and the system transitions to the next state $s_{t+1}$ and steps 1 - 3 are repeated.
5. If $\mathfrak{g}_c(s_t) = 1$:
   ○ $F^{\boldsymbol{\theta}}$ is computed using $s_t$ and the Generator action $\theta^c \sim g$.
   ○ Each agent $i \in \mathcal{N}$ receives a reward $r_i + F^{\boldsymbol{\theta}}$ and the system transitions to $s_{t+1}$.
6. At time $t + 1$ if the intrinsic reward terminates then steps 1 - 3 are repeated or if the intrinsic reward has not terminated then step 5 is repeated.

---

[2]More precisely, $\{\tau_k\}_{k \geq 0}$ are *stopping times* (Øksendal, 2003).

### 4.1 THE LEARNING PROCEDURE

In Sec. 5, we provide the convergence properties of the algorithm, and give the full code of the algorithm in Sec. 9 of the Appendix. The algorithm consists of the following procedures: the Generator updates its policy that determines the values $\theta$ at each state and the states to perform a switch while the agents $\{1, \ldots, N\}$ learn their individual policies $\{\pi_1, \ldots, \pi_N\}$. In our implementation, we used proximal policy optimization (PPO) (Schulman et al., 2017) as the learning algorithm for both the Generator's intervention policy $\mathfrak{g}_c$ and Generator's policy $g$. For the $N$ agents we used MAPPO (Yu et al., 2021). for the Generator $L$ term we use[3] $L(s_t, \boldsymbol{a}_t) := \|\hat{h} - h\|_2^2$ where $\hat{h}$ is a random initialised network which is the target network which is fixed and $h$ is the *prediction function* that is consecutively updated during training. We constructed $F^{\boldsymbol{\theta}}$ using a fixed neural network $f : \mathbb{R}^d \mapsto \mathbb{R}^m$ and a one-hot encoding of the action of the Generator. Specifically, $i(\theta_t^c)$ is a one-hot encoding of the action $\theta_t^c$ picked by the Generator. Thus, $F^{(\theta_t^c, \theta_{t-1}^c)} = i(\theta_t^c) - \gamma^{-1} i(\theta_{t-1}^c)$. The action set of the Generator is $\Theta \equiv \{1\}$ where $g$ is an MLP $g : \mathbb{R}^d \mapsto \mathbb{R}^m$. Extra details are in Sec. 9.

## 5 CONVERGENCE AND OPTIMALITY OF LIGS

We now show that LIGS converges and that the solution ensures a higher performing agent policies. The addition of the Generator's RL process which modifies $N$ agents' rewards during learning can produce convergence issues (Zinkevich et al., 2006). Also to ensure the framework is useful, we must verify that the solution of $\mathcal{G}$ corresponds to solving the MDP, $\mathfrak{M}$. To resolve these issues, we first study the stable point solutions of $\mathcal{G}$. Unlike MDPs, the existence of a solution in Markov policies is not guaranteed for MGs (Blackwell & Ferguson, 1968) and is rarely computable (except for special cases such as *team* and *zero-sum* MGs (Shoham & Leyton-Brown, 2008)). MGs also often have multiple stable points that can be inefficient (Mguni et al., 2019); in $\mathcal{G}$ such stable points would lead to a poor performing agent joint policy. We resolve these challenges with the following scheme:

**[I]** LIGS preserves the optimal solution of $\mathfrak{M}$.
**[II]** The MG induced by LIGS has a stable point which is the convergence point of MARL.
**[III]** LIGS yields a team payoff that is (weakly) greater than that from solving $\mathfrak{M}$ directly.
**[IV]** LIGS converges to the solution with a linear function approximators.

In what follows, we denote by $\boldsymbol{\Pi} := \times_{i \in \mathcal{N}} \Pi_i$. The results are built under Assumptions 1 - 7 (Sec. 15 of the Appendix) which are standard in RL and stochastic approximation theory. We now prove the result **[I]** which shows the solution to $\mathfrak{M}$ is preserved under the influence of LIGS:

**Proposition 1** *The following statements hold:*
*i)* $\max_{\boldsymbol{\pi} \in \boldsymbol{\Pi}} v^{\boldsymbol{\pi}, g}(s, \cdot) = \max_{\boldsymbol{\pi} \in \boldsymbol{\Pi}} v^{\boldsymbol{\pi}}(s), \ \forall s \in \mathcal{S}, \forall i \in \mathcal{N}, \forall g$ *where* $v^{\boldsymbol{\pi}}(s) = \mathbb{E}_{\boldsymbol{\pi}} \left[ \sum_{t=0}^{\infty} \gamma^t R \right]$.
*ii) The Generator's optimal policy maximises* $v^{\boldsymbol{\pi}}(s) = \mathbb{E} \left[ \sum_{t=0}^{\infty} \gamma^t R(s_t, \boldsymbol{a}_t) \right]$ *for any* $s \in \mathcal{S}$.

Result (i) says that the agents' problem is preserved under the Generator's influence. Moreover the agents' (expected) total return is that from the environment (extrinsic rewards). Result (ii) establishes that the Generator's optimal policy induces it to maximise the agents' joint (extrinsic) total return. The result is proven by a careful adaptation of the policy invariance result in Ng et al. (1999) to our MARL switching control setting where the intrinsic-reward is not added at all states. Building on Prop. 1, we deduce the following result:

**Corollary 1** *LIGS preserves the dec-MDP played by the agents. In particular, let* $(\hat{\boldsymbol{\pi}}, \hat{g})$ *be a stable point policy profile[4] of the MG induced by LIGS,* $\mathcal{G}$ *then* $\hat{\boldsymbol{\pi}}$ *is a solution to the dec-MDP,* $\mathfrak{M}$.

Therefore, the introduction of the Generator does not alter the fundamentals of the problem. Our next task is to prove the existence of a stable point of the MG induced by LIGS and show it is a limit point of a sequence of Bellman operations. To do this we prove that a stable solution of $\mathcal{G}$ exists and that $\mathcal{G}$ has a special property that permits its stable point to be found using dynamic programming. The following result establishes that the solution of the MG $\mathcal{G}$, can be computed using RL methods:

---

[3]This is similar to random network distillation (Burda et al., 2018) however the input is over the space $\mathcal{A} \times \mathcal{S}$.
[4]By stable point profile we mean a Markov perfect equilibrium (MPE) (Fudenberg & Tirole, 1991).

**Theorem 1** *Given a function $V : \mathcal{S} \times \mathcal{A} \to \mathbb{R}$, $\mathcal{G}$ has a stable point given by $\lim\limits_{k \to \infty} T^k V^{\boldsymbol{\pi}, g} = \sup\limits_{\hat{\boldsymbol{\pi}} \in \boldsymbol{\Pi}} V^{\boldsymbol{\pi}, \hat{g}} = V^{\boldsymbol{\pi}^\star, g^\star}$ where $(\boldsymbol{\pi}^\star, g)$ is a stable solution of $\mathcal{G}$ and $T$ is the Bellman operator (c.f. (5)).*

Theorem 1 proves that the MG $\mathcal{G}$ (which is the game that is induced when Generator plays with the $N$ agents) has a stable point which is the limit of a dynamic programming method. In particular, it proves the that the stable point of $\mathcal{G}$ is the limit point of the sequence $T^1 V, T^2 V, \dots,$. Crucially, (by Corollary 1) the limit point corresponds to the solution of the dec-MDP $\mathcal{M}$. Theorem 1 is proven by firstly proving that $\mathcal{G}$ has a dual representation as an MDP whose solution corresponds to the stable point of the MG. Theorem 1 enables us to tackle the problem of finding the solution to $\mathcal{G}$ using distributed learning methods i.e. MARL to solve $\mathcal{G}$. Moreover, Prop. 1 indicates by computing the stable point of $\mathcal{G}$ leads to a solution of $\mathfrak{M}$. These results combined prove **[II]**. Our next result characterises the Generator policy $\mathfrak{g}_c$ and the optimal times to activate $F$. The result yields a key aspect of our algorithm for executing the Generator activations of intrinsic rewards:

**Proposition 2** *The policy $\mathfrak{g}_c$ is given by: $\mathfrak{g}_c(s_t, I_t) = H(\mathcal{M}^{\boldsymbol{\pi}, g} V^{\boldsymbol{\pi}, g} - V^{\boldsymbol{\pi}, g})(s_t, I_t)$, $\forall (s_t, I_t) \in \mathcal{S} \times \{0, 1\}$, where $V^{\boldsymbol{\pi}, g}$ is the solution in Theorem 1, $\mathcal{M}$ is the Generator's intervention operator (c.f. (4)) and $H$ is the Heaviside function, moreover $\tau_k = \inf\{\tau > \tau_{k-1} | \mathcal{M}^{\boldsymbol{\pi}, g} V^{\boldsymbol{\pi}, g} = V^{\boldsymbol{\pi}, g}\}$.*

In general, introducing intrinsic rewards or shaping rewards may undermine learning and worsen overall performance. We now prove that the LIGS framework introduces an intrinsic reward which yields better performance for the $N$ agents as compared to solving $\mathfrak{M}$ directly (**[III]**).

**Theorem 2** *Each agent's expected return $v^{\boldsymbol{\pi}, g}$ whilst playing $\mathcal{G}$ is (weakly) higher than the expected return for $\mathfrak{M}$ (without the Generator) i.e. $v^{\boldsymbol{\pi}, g}(s, \cdot) \geq v^{\boldsymbol{\pi}}(s)$, $\forall s \in \mathcal{S}$, $\forall i \in \mathcal{N}$.*

Theorem 2 shows that the Generator's influence leads to an improvement in the system performance. Note that by Prop. 1, Theorem 2 compares the environment (extrinsic) rewards accrued by the agents so that the presence of the Generator increases the total expected environment rewards. We complete our analysis by extending Theorem 1 to capture (linear) function approximators which proves **[IV]**. We first define a *projection* $\Pi$ by: $\Pi\Lambda := \arg\min\limits_{\bar{\Lambda} \in \{\Phi r | r \in \mathbb{R}^p\}} \left\| \bar{\Lambda} - \Lambda \right\|$ for any function $\Lambda$.

**Theorem 3** *LIGS converges to the stable point of $\mathcal{G}$, moreover, given a set of linearly independent basis functions $\Phi = \{\phi_1, \dots, \phi_p\}$ with $\phi_k \in L_2, \forall k$. LIGS converges to a limit point $r^\star \in \mathbb{R}^p$ which is the unique solution to $\Pi\mathfrak{F}(\Phi r^\star) = \Phi r^\star$ where $\mathfrak{F}\Lambda := \hat{R} + \gamma P \max\{\mathcal{M}\Lambda, \Lambda\}$. Moreover, $r^\star$ satisfies: $\|\Phi r^\star - Q^\star\| \leq (1 - \gamma^2)^{-1/2} \|\Pi Q^\star - Q^\star\|$.*

The theorem establishes the convergence of LIGS to a stable point (of $\mathcal{G}$) with the use of linear function approximators. The second statement bounds the proximity of the convergence point by the smallest approximation error that can be achieved given the choice of basis functions.

## 6 EXPERIMENTS

We performed a series of experiments on the Level-based Foraging environment (Papoudakis et al., 2020) to test if LIGS: **1.** Efficiently promotes joint exploration **2.** Optimises convergence points by inducing coordination. **3.** Handles sparse reward environments. In all tasks, we compared the performance of LIGS against MAPPO (Yu et al., 2021), QMIX (Rashid et al., 2018); intrinsic reward MARL algorithms LIIR (Du et al., 2019), LICA (Zhou et al., 2020a), and a leading MARL exploration algorithm MAVEN (Mahajan et al., 2019). We then compared LIGS against these baselines in StarCraft Micromanagement II (SMAC) (Samvelyan et al., 2019). Lastly, we ran a detailed suite of ablation studies (see Appendix) in which we demonstrated LIGS' flexibility to accommodate i) different MARL learners, ii) different $L$ bonus terms for the Generator objective. We also demonstrated the necessity of the switching control component in LIGS and LIGS' improved use of exploration bonuses.

## 6.1 COOPERATIVE FORAGING TASKS

**Experiment 1: Coordinated exploration.** We tested our first claim that LIGS promotes coordinated exploration among agents. To investigate this, we used a version of the level-based foraging environment (Papoudakis et al., 2020) as follows: there are $n$ agents each with level $a_i$. Moreover, there are 3 apples with level $K$ such that $\sum_{i=1}^{N} a_i = K$. The only way to collect the reward is if all agents collectively enact the collect action when they are beside an apple. This is a challenging joint-exploration problem since to obtain the reward, the agents must collectively explore joint actions across the state space (rapidly) to discover that simultaneously executing collect near an apple produces rewards. To increase the difficulty, we added a penalty for the agents failing to coordinate in collecting the apples. For example, if only one agent uses the collect action near an apple, it gets a negative reward. This results in a non-monotonic reward structure. The performance curves are given in Fig. 2 which shows LIGS demonstrates superior performance over the baselines.

**Experiment 2: Optimal joint policies.** We next tested our second claim that LIGS can promote convergence to joint policies that achieve higher system rewards. To do this, we constructed a challenging experiment in which the agents must avoid converging to suboptimal policies that deliver positive but low rewards. In this experiment, the grid is divided horizontally in three sections; top, middle and bottom. All grid locations in the top section give a small reward $r/n$ to the agent visiting them where $n$ is the number of tiles in the each section. The middle section does not give any rewards. The bottom section rewards the agents depending on their relative positions. If one agent is at the top and the other at the bottom, the agent at the bottom receives a reward $-r/n$ each time the other agent receives a reward. If both agents are at the bottom, then one of the tiles in this section will give a reward $R, r/2 < R < r$ to both agents. The bottom section gives no reward otherwise. The agents start in the middle section and as soon as they cross to one section they cannot return to the middle. As is shown in Fig. 2, LIGS learns to acquire rewards rapidly in comparison to the baselines with MAPPO requiring around 400k episodes to match the rewards produced by LIGS.

**Experiment 3: Sparse rewards.** We tested our claim that LIGS can promote learning in MAS with sparse rewards. We simulate a sparse reward setting using a competitive game between two teams of agents. One team is controlled by LIGS while the other actions of the agents belonging to the other team are determined by a fixed policy. The goal is to collect the apple faster than the opposing team. Collecting the apple results in a reward of 1, and rewards are 0 otherwise. This is a challenging sparse reward since informative reward signals occur only apple when the apple is collected. As is shown in Fig. 1 both LIGS and MAPPO perform well on the sparse rewards environment, whilst the other baselines are all unable to learn any behaviour on this environment.

**Investigations.** We investigated the workings of the LIGS framework. We studied the locations where the Generator added intrinsic rewards in Experiments 1 and 2. As shown in the heatmap visualisation in Fig. 3, for Experiment 2, we observe that the Generator learns to add intrinsic rewards that guide the agents towards the optimal reward (bottom right) and away from the suboptimal rewards at the top (where some other baselines converge). This supports our claim that LIGS learns to guide the agents towards jointly optimal policies. Also, as Fig. 3 shows, LIGS's switching mechanism

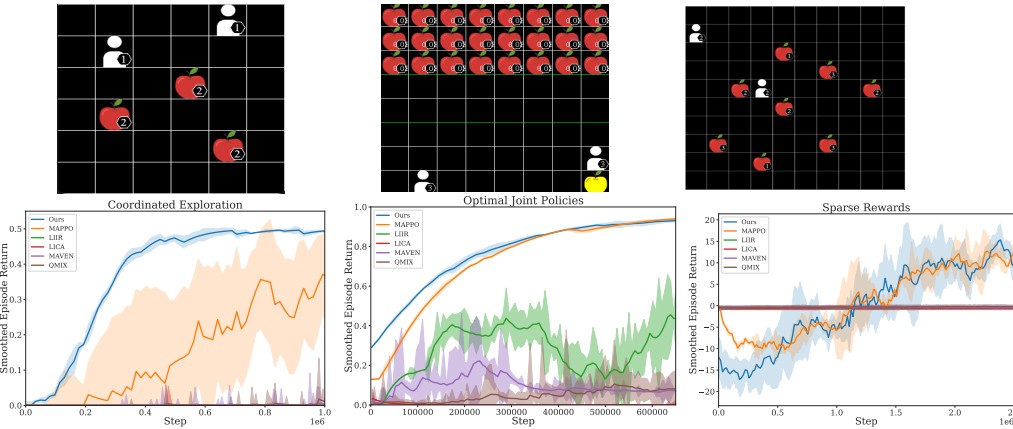

Figure 2: *Left.* Coordinated Exploration. *Centre.* Optimal joint policies. *Right.* Sparse rewards.

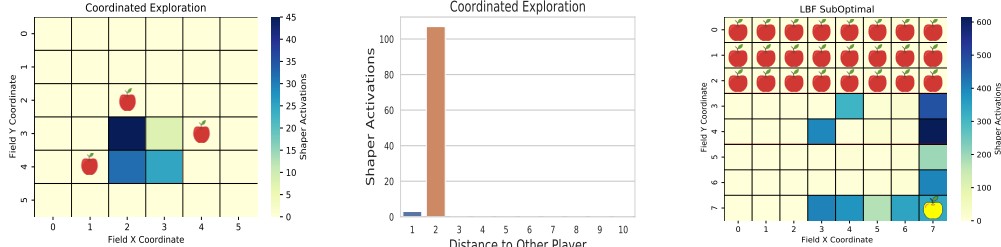

Figure 3: *Left.* Heatmap of Exp. 1 showing where Generator adds rewards. *Centre.* Plot of distance to other agent when Generator activates rewards in Exp 1. *Right.* Corresponding heatmap for Exp. 2.

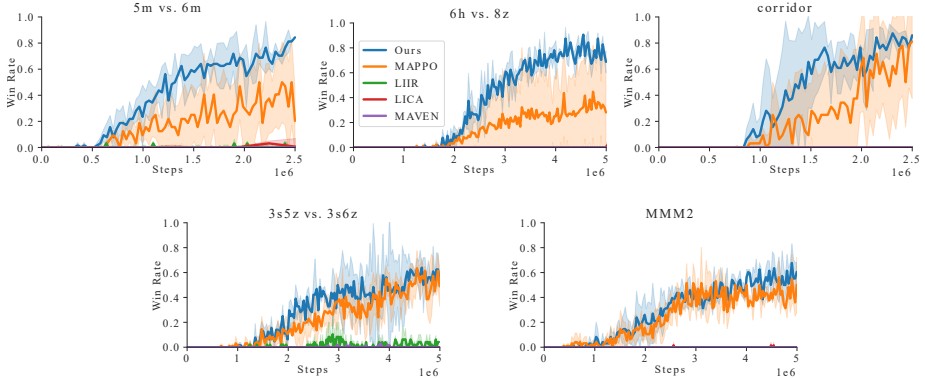

Figure 4: *Median win rate over the course of learning on SMAC.* LIGS outperforms the baselines on all maps. LIIR, LICA, and MAVEN are generally not visible as their win rate is negligible.

means that the Generator only adds intrinsic rewards at the most useful locations for guiding the agents towards their target. For Experiment 1, Fig. 3 shows that the Generator learns to guide the agents towards the apple which delivers the high rewards. Fig. 3 (Right) demonstrates a striking behaviour of the LIGS framework - it only activates the intrinsic rewards around the apple when *both* agents are at most 2 cells away from the apple. Since the agents receive positive rewards only when they arrive at the apple simultaneously, this ensures the agents are encouraged to coordinate their arrival and receive the maximal rewards and avoids encouraging arrivals that lead to penalties.

## 6.2 LEARNING PERFORMANCE IN STARCRAFT MULTI-AGENT CHALLENGE

To ascertain if LIGS is effective even in complex environments, we ran it on on the following SMAC maps *5m vs. 6m* (hard), *6h vs. 8z*, *Corridor*, *3s5z vs 3s6z* and *MMM2* (super hard). These maps vary in a range of MARL attributes such as number of units to control, environment reward density, unit action sets, and (partial)-observability. In Fig. 4, we report our results showing 'Win Rate' vs 'Steps'. These curves are generated by computing the median win rate (vs the opponent) of the agent at regular intervals during learning. We ran 3 seeds of each algorithm (further setup details are in the Supp. material Sec 14). LIGS outperforms the baselines in all maps. In *5m vs. 6m* and *6h vs. 8z*, the baselines do not approach the performance of LIGS. In *Corridor* MAPPO requires over an extra million steps to match LIGS. In *3s5z vs. 3s6z* and *MMM2*, LIGS still outperforms the baselines. In summary, LIGS shows performance gains over all baselines in SMAC maps which encompass diverse MAS attributes.

## 7 CONCLUSION

We introduced LIGS, a novel framework for generating intrinsic rewards which significantly boosts performance of MARL algorithms. Central to LIGS is a powerful adaptive learning mechanism that generates intrinsic rewards according to the task and the MARL learners' joint behaviour. Our experiments show LIGS induces superior performance in MARL algorithms in a range of tasks.

## 8 ACKNOWLEDGEMENTS

We would like to thank Matthew Taylor and Aivar Sootla for their helpful comments.

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
