# OpenReview forum: "LIGS: Learnable Intrinsic-Reward Generation Selection for Multi-Agent Learning "
_ICLR.cc/2022/Conference — ICLR 2022 Poster_

### Official Review · Reviewer_crmy · 2021-11-02

**Correctness:** 3
**Technical Novelty And Significance:** 3
**Empirical Novelty And Significance:** 3
**Recommendation:** 5
**Confidence:** 4

**Details Of Ethics Concerns:**

The reviewer does not see obvious ethics concerns.

**Main Review:**

As fas as the reviewer is concerned, the proposed method is novel. However, there are some drawbacks that can be improved.

***Soundness***

The objective of Generator contains four terms: (1) environmental rewards, (2) learned intrinsic rewards, (3) rewarding costs, and (4) a RND-style exploration bonus. Learning incentivizing policies of Generator may not be easier than learning without it because the Generator faces a large space that has a similar size of joint action-observation space.

Some claims need more consideration. For example, the authors hold that "CT-DE methods are however, prone to convergence to suboptimal joint policies (Mahajan et al., 2019). ... In such situations, (random) occurrences of successful coordination are improbable, moreover value factorisations, e.g. QMIX (Rashid et al., 2018), cannot represent non-monotonic team rewards." Not all CTDE methods have this problem. QTRAN, QTRAN++, weighted QMIX, and QPLEX can represent non-monotonic Qs and are free from the issue in Mahajan's paper.  The reviewer recommend rewriting the third paragraph of the Introduction section to better motivate the proposed method

Discussion of related works should be improved. For example, there are many multi-agent exploration methods expect for MAVEN, and previous works on learning (or designing) multi-agent intrinsic rewards (e.g., using transformers or curiosity-based intrinsic rewards) should be discussed.

***Evaluation***

The reviewer was expecting an ablation study regarding the exploration bonus. Previous work has shown that RND works well on SMAC. The reviewer would increase the score if the authors can prove that LIGS performs relatively well without the RND-like objective.



**Summary Of The Paper:**

The paper focuses on learning intrinsic rewards for multi-agent reinforcement learning, which is an important problem. Different from previous works on this topic, the authors propose to train an agent with a learnable gating function that incentives other agents. Theoretical analysis and empirical evaluation are provided to prove the effectiveness of the proposed method.

**Summary Of The Review:**

The reviewer finds the idea novel and the method solid. However, the empirical evaluation, the discussion of related works, and the presentation of the paper can be further improved.

---

> ### Author Response · Authors · 2021-11-23
> **Authors' response**
>
> We thank the reviewer for their careful reading and valuable comments. We have responded to the reviewer's comments individually.
>
> **Performance without the RND-like objective**
>
> **A.** Thanks for the valuable suggestion. We have included experimental results which confirm LIGS performs very well without the RND term (and in fact entirely comparably with a very simple count-based bonus instead of RND) in Generator's objective as requsted - ***please see Section 11***. We also demonstrated that our method significantly outperforms leading MARL learners with an RND intrinsic reward included in their objective. This shows that LIGS is able to significantly augment the benefits of applying RND directly to the agents' objectives - ***please see Section 12***.
>
>
> **"Learning incentivizing policies [...] may not be easier [...] because the Generator faces a large space that has a similar size of joint action-observation space."**
>
> **A.** Key to our framework is the switching control framework. It reduces the set of states that Generator learns intrinsic rewards - note that decision space for determining to add intrinsic rewards is binary $\\{0,1\\}$. In our experiments, we chose the size of the action set for Generator is a singleton which means that the decision space for Generator is $|\mathcal{S}|\times \\{0,1\\}$ in contrast to the decision space for a MARL agent algorithm which is $|\mathcal{S}||\mathcal{A}|$.
>
> **Discussion on CT-DE methods \& related work**
>
> **A.** We sincerely thank the reviewer for their valuable comments. We have adjusted these paragraphs accordingly. Please see the updated version of the paper.
>
> **Other further improvements to empirical evaluation**
>
> Further to the reviewer's suggestion about improving the empirical evaluation, we have now also added a detailed investigation of the behaviour of LIGS. Please see Pg. 9 and Fig. 2 in our updated version. We have included a study of the set of states for which LIGS learns to add intrinsic rewards and the conditions under which intrinsic rewards are added. In this study we show that LIGS adds rewards at states that lead to it jointly guiding the agents towards optimal goals and encourages coordinated behaviour. We also show that the switching mechanism switches on only at the most relevant states to achieve the above and, when required, switches off to carefully avoid encouraging miscoordination and the associated penalties.

---

### Official Review · Reviewer_nTCC · 2021-11-02

**Correctness:** 4
**Technical Novelty And Significance:** 4
**Empirical Novelty And Significance:** 4
**Recommendation:** 8
**Confidence:** 3

**Main Review:**

The paper is well written and relatively easy to follow, although the notation is at time heavy. This reviewer noted the extensive support material, including relevant material for reproducing the experiments, an ablation study and further the mathematical derivations to support the work. I could not go through section 14 containing the proofs of the technical results, which are several pages of extensive mathematical notation. So I cannot comment on the correctness and I wonder whether such an extensive material is really essential, although as part of the support material does not affect the fruition of the main paper.

[General readability] In section 3, paragraph 1, "the system is in a state s and each agent takes an action a”. The concept of “system” is not defined here. At first, I would assume that it is one agent, and not a system, that can be in a state. Similarly, as I came to the concept of joint-action, it was not clear to me as I was under the impression that each agent makes one individual choice. Also unclear is how a joint action can produce a reward for one agent, and lead to one transition since a I previously assumed a transition to defined for a state-action pair, and not a state-multiple-actions pair. This lack of clarity might be due to my limited familiarity of this very particular setup, but I feel that other readers in the area of RL could benefit from a slightly more explicit explanation on this point.


My main observation is that more explanations would be beneficial in relation to  how the properties of the algorithm described in section 6 relate to the experimental evidence in section 7. In particular, what specific aspects of the simulations complement the theoretical discussion above, and in which way? Are the experiment meant to assess specific metrics of interest, to provide a quantitative advantage over existing SOTA algorithms in the proposed benchmarks? Or else? While the narrative appears to assume that the answer to these questions are self-evident, I would argue that more justifications on the precise aims of both theoretical and experimental evidence would strengthen the paper.

I would have liked to read more about the limitations and drawback of the proposed approach in relation to existing algorithms, in particular those used for comparison. While performance is improved, what is the price to pay for that, if any? Are there particular implications from a computational perspective that need to be considered? Considerations on tuning hyper-parameters, or other elements that could help appreciate possible limitations?




**Summary Of The Paper:**

The paper describes a novel reinforcement learning algorithm for multi-agent system (MARL) that employs a generator of intrinsic reward and a switching control system that helps to regulate intrinsic control. Crucially, the intrinsic reward is learned to better fit the particular task being learned. The paper claims that the proposed algorithm helps with exploration as well as preservation of known policies. The paper has a strong theoretical background with a section that illustrates the properties of convergence and optimality. The experimental results appear to justify the approach with superior performance with respect to the baselines. The paper deals with an emerging and interesting area of RL and proposes a new mechanism for co-ordinated RL agents.


**Summary Of The Review:**

This is a solid paper that proposed a promising approach to MARL. Many details in the paper require extensive examination, care, and previous knowledge. The paper is fairly notation-heavy, but it makes a fair effort to explain concepts also in plain English. The results are promising and I believe it can provide a valid contribution to ICLR.

---

> ### Author Response · Authors · 2021-11-24
> **Authors' response**
>
> We thank the reviewer for their careful reading and valuable comments.
>
> We hope the reviewer will share their enthusiasm for the paper with other reviewers.
>
> **Specific aspects of the simulations that complement our theory and justifications of the precise aims**
>
> We thank the reviewer for these two insightful comments.  We have now included a series of studies within the paper to deepen the connection between the claims, theoretical results, and the aims and benefits of the LIGS framework.
>
> Specifically:
>
> *To evaluate the switching control aspect discussed in Sec. 4 and the result of Prop. 2:* we have included a study of the set of states for which LIGS learns to add intrinsic rewards and the conditions under which intrinsic rewards are added. In this study, we show that the switching mechanism switches on only at the most relevant states to achieve the improved joint outcomes for the agents' performance in accordance with Prop. 2 and, when required, switches off to carefully avoid encouraging miscoordination and the associated penalties. We show here that LIGS adds rewards at states that lead to it jointly guiding the agents towards optimal goals and encourages coordinated behaviour in accordance with Theorem 2.
>
> *To evaluate the benefits of using the LIGS framework versus applying an intrinsic bonus term e.g. RND (as discussed in Sec. 1):*  we included a new study that demonstrates that LIGS significantly outperforms leading MARL learners with an RND intrinsic reward included in their objective. This shows that LIGS is able to significantly augment the benefits of applying RND directly to the agents' objectives - please see Sec. 12.
>
> *To evaluate the impact of (parameter) choices on the framework:* we also included an Ablation study to demonstrate that LIGS is flexible in its ability to accommodate different bonus terms in Generator's objective, please see Sec. 11.

---

> > ### Comment · Reviewer_nTCC · 2021-11-29
> > **Response**
> >
> > I'm happy to see further improvements in response to my observations. I confirm my assessment that this is a good paper well above the acceptance threshold.

---

> > > ### Author Response · Authors · 2021-11-29
> > > **Re: Response**
> > >
> > > We are grateful to the reviewer for their appreciation of our paper, kind words and their suggestions which have further improved the work.
> > >
> > > We have not yet had any activity from other reviewers following our response and updates (but look forward to it).
> > >
> > > We hope that the reviewer would kindly voice their appreciation during the internal discussion phase with other reviewers + ACs so that the work can be published & shared with the community.

---

### Official Review · Reviewer_W87Z · 2021-11-03

**Correctness:** 3
**Technical Novelty And Significance:** 3
**Empirical Novelty And Significance:** Not applicable
**Recommendation:** 6
**Confidence:** 4

**Main Review:**

The first thing that needs to be said about this paper is that it is hard to read. There are several elements that lead to this overall conclusion:
- There are many small errors, inconsistencies, and unnecessary elements in the text and equations.
- The presentation of the contribution is smeared out over several pages, instead of providing one clear overview followed by further details where necessary.
- The paper needs some language editing.

Examples of the small issues with the text (this is not meant to be an exhaustive list, I would ask the authors to critically re-read and edit their work):
- Section 3, second line: does the set $\mathcal{N}$ contains the indices $1$ through $N$, or the agents? It seems that in most places where you refer to $\mathcal{N}$, the sentence would be just as clear without it.
- Section 7.2: you promise four SMAC maps, and then list (and show results on) only three
- Algorithm 2:
  - Line 1: what is $\pi_0$? Is it the set of agent policies {$\pi^i_0$}?
  - Line 3 is informative, please have a similar line for the second loop starting on line 16.
  - Lines 8-9, 15, 18: are $g_t$ and $q_t$ the same?
  - Input: the parameters of $h$ and $\hat{h}$ are probably $\theta_h$ and $\theta_\hat{h}$
  - Lines 18-21: no need for control flow here, that's taken care of by $r_t^i$ being zero if $q_t$ is zero itself.
  - What happens on line 27?
  - Section 4.2, which is supposed to show the learning algorithm (PPO), does not seem to be present in the paper. Section 4, for which the same claim is made, does not introduce PPO.
- Page 18 (section / Appendix 14): in the long derivation, there are inconsistencies in how $t$ is shifted to $t+1$. It is also not clear to me where the the last term on the second line comes from. In the second half of that derivation, a capital $K$ appears, without it being made clear what it means. As it stands, I'm not convinced of the correctness of that derivation.

In spite of the errors and inconsistencies, I found Algorithm 2 to be the best presentation of the paper’s contribution. Please consider replacing sections 3-5 in the main text with a corrected version of Algorithm 2.

All that being said, the results do look promising, and the method is interesting. Some more detailed comments and requests:
- While the results look good, MAPPO isn’t quite state of the art. Please compare to something that beats MAPPO, e.g. EMC, https://openreview.net/forum?id=cLYyCXHU7g1n.
- The paper's contribution could be more significant if a bit more time/space was spent on the results (this should not be a problem if the presentation of the method is condensed). In particular investigating how the method works in practice and in that way answering the question why it actually works would be very valuable. Suggested questions to answer (this might be done with the gridworld environments):
  - What does the exploration reward generator actually learn? What does it reward?
  - What does the gate / switching control mechanism learn? When does it switch on and off?

Other comments:
- ‘Novel framework’ sounds a bit grand. The centralized exploration shaping reward that is introduced looks like a good idea, but it’s not a novel framework.
- Central coordination of exploration, like the method here is doing, reduces the independence of the agents. Centralized training in general does that too, though, and there are no gradients between the agents being introduced here, so it seems okay. But it’s good to clearly position the contribution in this respect; the method does further reduce the independence, and thereby the multi-agent aspect of MARL.
- The fact that the gate is binary seems targeted to environments like the gridworld. Have you experimented with a continuous gate, or no gate at all? What are the states in the SMAC environments where it switches on? More generally, what are the assumptions that make the switching work?


**Summary Of The Paper:**

This paper introduces a learned centralized exploration reward for multi-agent settings. The exploration reward is factorized into an on/off gate (dubbed ‘switching control’) and a scale function. Some mathematical derivations are included (sketched in the main text, with details in the appendices) to provide theoretical guarantees on how the exploration reward changes the solutions the training procedure might find. Evaluations on gridworld environments that target specific difficulties of multi-agent exploration show promising results. Some maps from the SMAC benchmark are also included, and again show good results.

**Summary Of The Review:**

While the results look good, the text and equations are unclear. My current recommendation is to reject the paper, because the presentation quality falls short of the expected standard. However, there are interesting aspects to the proposed method, and the results look promising. If the presentation is improved significantly, I would be open to recommending a weak accept. I would strongly recommend to the authors to condense the presentation of the method in the main text and spend more time on evaluating the method. With more evaluation and clarification of how the method functions in practice, and if the further results are also good, I would consider raising my recommendation to accept.

EDIT: the resubmission addresses a significant portion of my readability concerns, I'm raising my score to a weak accept.

---

> ### Author Response · Authors · 2021-11-23
> **Authors' response**
>
> We thank the reviewer for their careful reading and valuable comments. We have responded to the reviewer's comments individually.
>
> **Readability**
>
> **A.** Thanks for the comments, we have made various adjustments to the structure to improve the flow and readability as per the reviewer's comments (for example Sec. 4 has a more linear flow). We have also given the paper a thorough polish to eliminate small typos. Please see the updated version. We hope that the reviewer agrees with the other reviewers that we have also taken much care to explain the various concepts that appear in the paper and that audiences/reviewers who seek strong theoretical grounding may prefer an analytic presentation style.
>
> **What does the generator actually learn and reward, when does the switching control mechanism switch on/off?**
>
> **A.** We thank the reviewer for these very valuable suggestions. We have now added a detailed investigation of the behaviour of LIGS. Please see Pg. 9 and Fig. 2 in our updated version. We have included a study of the set of states for which LIGS learns to add intrinsic rewards and the conditions under which intrinsic rewards are added.  In this study we show that LIGS adds rewards at states that lead to it jointly guiding the agents towards optimal goals and encourages coordinated behaviour. We also show that the switching mechanism switches on only at the most relevant states to achieve the above and, when required, switches off to carefully avoid encouraging miscoordination and the associated penalties. We have also added an additional study of LIGS against exploratory intrinsic reward methods, please see Sec. 12.
>
>
> **Explanation of the derivation.**
>
> **A.** The shift from $t$ to $t+1$ is accounted for in the change in the bounds of the summation. Regarding the last term, we just isolated the first term in the summation (NB the summations starts at $\tau_1+1$ not $\tau_1$). We added an extra line in the updated version the to make this even clearer. With this the reviewer can verify the correctness of the proof.
>
> **The term $K$.**
>
> **A.** This is stated in Assumption 6 (referred to in the proof) where we state that $K$ denotes the (finite) total number of intrinsic reward activations (interventions). We updated the script to help point this out (please see our updated version).
>
> **Details of Algorithm 2:**
>
> **A.**  Thanks for pointing out the small number of typos (we have updated accordingly). Please find the responses to your points about Algorithm 2 (in order):
>
> * Yes, $\pi_0$ is the set of initial agent policies (please now see the first line).
> * We will insert a brief description for the loop on Line 16.
> * Yes, they perform the same function in the sense that they mediate the reward the agent sees depending on if the switch to add intrinsic reward is on or off ($g_t, q_t = 1$ or $0$, resp.).
> * Thanks, we have replaced $\theta_{\hat{f}}$ with $\theta_h$ and $\theta_{\hat{h}}$.
> * Your point of view is correct, we just added the control flow for extra clarity.
> * Line 27 is just multiplying the switch activations $g_t$ with the actual switching cost defined in the inputs to the algorithm.
> * For the PPO update present in the Algorithm in Sec. 8, we refer to the reference to the Proximal Policy Optimization algorithm stated in Sec 4.1.
>
> **Please compare to something that beats MAPPO, e.g. EMC.**
>
> **A.** Many thanks for the reference. We were not able to find the code for this very recent work the reviewer has cited. However, we have so far cited this paper and include it in the camera-ready should the paper be accepted. Please also note the plug \& play aspect of our framework (see Sec. 9).
>
> **The binary aspect seems targeted to environments like the gridworld.**
>
> **A.** Although the switch itself is a binary mechanism, it can work on various environments. To be clear, the binary gate simply determines whether an intrinsic reward should be added at a particular region or state. For example the state space in our StarCraft II (SMAC) experiments is in fact continuous while in the Foraging environment it is discrete.
>
> **Assumptions that make the switching work?**
>
> **A.** In our theory, we make the assumption that the number of switches is finite (see Assumption 6).
>
> **Time spent on evaluating the method and more evaluation and clarification of how the method works**
>
> **A.** Thanks again to the reviewer for their suggestions. We have now added detailed evaluation analyses of the method to the paper, namely the evaluations requested by the reviewer and, two further Ablation studies in Secs. 11 and 12. To maintain a theoretically grounded approach we included a significant amount of theory to fully support our approach. We hope the reviewer agrees that we have now provided a solid evaluation of the method for practitioners while appreciating that some audiences who seek theoretical grounding may also desire an analytic presentation.

---

### Official Review · Reviewer_vU2u · 2021-11-08

**Correctness:** 3
**Technical Novelty And Significance:** 3
**Empirical Novelty And Significance:** 2
**Recommendation:** 5
**Confidence:** 4

**Main Review:**

From the perspective of learning intrinsic rewards and preserving the optimality, instead of hand-craft intrinsic rewards, it is a worthwhile investigation direction.

However, there are some confusions about the method and experiments. If all these concerns are addressed well by the authors, I am willing to increase the score.

First, I think the challenges mentioned in Section 1 are redundant. The first two points can be concluded into one point.

Second, in common, each agent should have its own reward function R_i(o_i,a_i), not from the global reward function. Only if each agent receives the team reward at each step, each agent’s objective is to maximize the team reward. However, in this paper, each agent maximizes its own expected return. Especially, this paper assumes some setting that each agent’s reward is non-monotonic to the team reward. Therefore, it is not suitable to obtain an individual reward from the global reward function.

Third, about the switch controls, I am confused about equation 2, where it uses I_t, while how to calculate or define I_t is not described. Instead, I_{\tau_{k+1}} is defined as the reverse value of I_{\tau_{k}}. Does this mean at each step, if I_t is 1, the value of I_{t+1} at the next state will switch to 0, and vise versa? Why is this changing flow? How does it find important states? And as a consequence, why equation 2 can be reformulated as equation 3 is confusing.

What does ” Generator constructs intrinsic rewards that are tailored for the specific setting” mean?

Finally, about the experiments, since I have a rich understanding of starcraft II, I am concerned that the selected maps in starcraft II may not well support the method. In these maps, just simply increase the exploration rate, many MARL algorithms can achieve better results than that shown in this paper, which can also be found in [1].

There exists some typos throughout the paper. So I recommend authors repolish the paper to make it clearer.
Such as, “with bilevel approach” ->” with a bilevel approach”
“Generator’s objective is:” repeats twice.
“(potentially away from suboptimal trajectories, c.f. Experiment 2) and enable”  -> enables

[1] Revisiting the Monotonicity Constraint in Cooperative Multi-Agent Reinforcement Learning. 2021


**Summary Of The Paper:**

This paper proposes a method, Learnable Intrinsic-reward Generation Selection (LIGS) to improve coordinated exploration. LIGS incorporates an extra agent, called Generator to learn what state to give what intrinsic reward for each agent. The intrinsic reward is potential-based, so it preserves the optimality. Experimental results on several domains show its advantages over several MARL methods.


**Summary Of The Review:**

From the perspective of learning intrinsic rewards and preserving the optimality, instead of hand-craft intrinsic rewards, it is a worthwhile investigation direction.

However, there are some concerns about the method and experiments needed to be addressed. So I give a borderline reject currently. If all these concerns are addressed well by the authors, I am willing to increase the score.

---

> ### Author Response · Authors · 2021-11-23
> **Authors' response**
>
> We thank the reviewer for their valuable comments and careful reading. We have responded to the reviewer's points individually.
>
> **Challenges in Sec. 1.**
>
> **A.** Thanks a lot for the suggestions, we have now merged the bullet points and made other amendments. Please see the revised version.
>
> **Equation 2 and the switch controls.**
>
> **A.** The reviewer has one or two misconceptions. Eqn. 2 describes each  objective for each agents 1,…N - these objectives include the intrinsic reward $F$. Eqn. 3 is Generator's objective (it is not a reformulation of Eqn. 2).
>
> When Generator activates the intrinsic rewards $F$ using its policy $\frak{g}_c$, the times of these activations are denoted by $\tau_1, \tau_3,\tau_5, \ldots$. The intrinsic rewards are deactivated probabilitistically at times $\tau_2, \tau_4,\ldots$, so that, as we explain on Pg 5, $\tau_1=4$ and $\tau_2 =8$, means that the intrinsic rewards are activated at times steps $t=4,5,6,7$.
>
> Regarding Eqn. 2, please firstly note the expression on $I$ (which is the coefficient on $F$) is $I_{\tau_{k+1}}=1-I_{\tau_{k}}$ - involving the switching times not $I_{t+1}=1- I_t$ as the reviewer has written. Since $I_{\tau_0}=0$, it should now be clear that $I$ in Eqn. 2 simply expresses that $F$ is activated at the points $\\{\tau_{2k+1}\\}$ (here $I=1$) and continues to be added until points $\\{\tau_{2k}\\}|_{k>0}$ (here $I=1-1=0$) as described above.
>
> **Meaning of "Generator constructs intrinsic rewards that are tailored for the specific setting".**
>
> **A.** In general our framework produces different intrinsic rewards for different environments. This follows from Generator choosing the magnitude of the intrinsic rewards and the states to add these rewards to using observations drawn from the specific environment. We do not use fixed hard-coded intrinsic rewards unlike many other intrinsic reward approaches. Please also see the newly added paragraph 2 on Pg. 9 which demonstrates this point.
>
> **""The selected maps in Starcraft II may not well support the method"".**
>
> **A.** Thanks for the comment. We have now include 2 further SMAC II maps to further show that our method improves performance in a diverse range of environments. We have also added the reference the reviewer has suggested. Please see Fig. 5 in Sec. 10 of the Supplementary material. We picked  maps which are defined as being *super-hard* or *hard*. In the SMAC maps in the paper, other baselines were not competitive against either MAPPO or LIGS. In Sec. 10 we therefore presented results that show that LIGS yields performance gains against the most competitive MARL baseline, MAPPO (notably leading to a $50\\%$ gain in win rate over MAPPO in MMM2 over MAPPO).
>
> **Reward functions**
>
> **A.** We consider a fully cooperative setting modelled by a dec-MDP. For our formulation, we use the same reward function as in various fully cooperative MARL/dec-MDP frameworks e.g. COMA (Foerster et. al, 2017), MAVEN (Mahajan et. al, 2020) and others. Here, the agents share the same reward function $R(s,\boldsymbol{a})$ (the reward function is dependent on the state not the observation). In our setting, each agent receives noisy feedback of their rewards hence we introduce $r_i\sim R$.
>
> **Typos**
>
> **A.** Thanks to the reviewer for pointing out the typos. We have now polished the paper and made various amendments to the text.

---

### Author Response · Authors · 2021-11-26
**Response to ALL REVIEWERS**

Dear Reviewers

Thank you for your careful reading of our paper and your insightful comments. **We have now added all the additional material requested by the reviewers for you to raise your scores to Accept, as you'd mentioned**. We have also put our best efforts into giving the paper a thorough polish and clarifying the answers to your questions. This includes:

•	**As requested by Reviewers W87Z, nTCC:** we added a study of where LIGS learns to add intrinsic rewards using its switching controls. This demonstrated that the switching mechanism activates only at the most relevant states to achieve the improved joint outcomes (see heatmaps in Fig.2). It also demonstrated that LIGS accounts for the agents’ relative positions when adding intrinsic rewards and a quite striking property that when required, LIGS switches off the intrinsic rewards to carefully avoid encouraging miscoordination. Please see Pg. 9, Fig. 2.

•	**As requested by Reviewer crmy:** we added a study of the impact of (parameter) choices on the framework. Specifically, we showed that LIGS accommodates different bonus terms in Generator's objective and performs well even when $L$ is a rudimentary count-based bonus.

•	**Further to the request by Reviewer crmy:** we also added a study of the benefit of the LIGS framework versus applying an intrinsic bonus term i.e. RND directly in the agents’ objectives. This demonstrated that LIGS significantly outperforms leading MARL learners with an RND intrinsic reward included in their objective. Please see Sec. 12.

•	**As requested by Reviewer vU2u:** we added several more maps of Starcraft and showed that our method outperforms leading MARL algorithms in all the added range of maps. Please see Fig. 5 in Sec. 10.

•	**As requested by Reviewers W87Z, nTCC:** we have now improved the presentation and structure of the paper.

We have also written out explanations to explain some of the small misconceptions that some reviewers had.

We would appreciate if the reviewers may take the time to review our answers and updates to the paper to confirm that they are now satisfied that their requests have all been met.

---

### Decision · Program_Chairs · 2022-01-20

**Decision:**

Accept (Poster)

**Comment:**

The paper addresses coordination improvement in the MARL setting by learning intristic rewards that motivate the exploration and coordination. The  paper is theoretically founded and the empirical evaluations back up the claims.

During the rebuttal the carried out an impressive amount of work. They provided several additional studies and substantially improved the presentation, addressing all of the reviewers' requests. Although not all the reviewers responded to the authors, the authors' response was taken into the account when recommending the decision.

Minor:
- The authors should comment on the learning intristic rewards with evolution (Faust et al, 2019): https://arxiv.org/abs/1905.07628